# Analysis of G-Quadruplex-Forming Sequences in Drought Stress-Responsive Genes, and Synthesis Genes of Phenolic Compounds in *Arabidopsis thaliana*

**DOI:** 10.3390/life13010199

**Published:** 2023-01-10

**Authors:** Petr Pečinka, Natália Bohálová, Adriana Volná, Kristýna Kundrátová, Václav Brázda, Martin Bartas

**Affiliations:** 1Department of Biology and Ecology, University of Ostrava, 70833 Ostrava, Czech Republic; 2Department of Biophysical Chemistry and Molecular Oncology, Institute of Biophysics of the Czech Academy of Sciences, 61265 Brno, Czech Republic; 3Department of Physics, University of Ostrava, 70833 Ostrava, Czech Republic

**Keywords:** G-quadruplex, PQS, *Arabidopsis thaliana*, drought stress, phenolic compounds

## Abstract

Sequences of nucleic acids with the potential to form four-stranded G-quadruplex structures are intensively studied mainly in the context of human diseases, pathogens, or extremophile organisms; nonetheless, the knowledge about their occurrence and putative role in plants is still limited. This work is focused on G-quadruplex-forming sites in two gene sets of interest: drought stress-responsive genes, and genes related to the production/biosynthesis of phenolic compounds in the model plant organism *Arabidopsis thaliana*. In addition, 20 housekeeping genes were analyzed as well, where the constitutive gene expression was expected (with no need for precise regulation depending on internal or external factors). The results have shown that none of the tested gene sets differed significantly in the content of G-quadruplex-forming sites, however, the highest frequency of G-quadruplex-forming sites was found in the 5′-UTR regions of phenolic compounds’ biosynthesis genes, which indicates the possibility of their regulation at the mRNA level. In addition, mainly within the introns and 1000 bp flanks downstream gene regions, G-quadruplex-forming sites were highly underrepresented. Finally, cluster analysis allowed us to observe similarities between particular genes in terms of their PQS characteristics. We believe that the original approach used in this study may become useful for further and more comprehensive bioinformatic studies in the field of G-quadruplex genomics.

## 1. Introduction

While the majority of the DNA in genomes is organized into a Watson–Crick B-DNA structure [1], the presence of various different structures with functional properties has been recently proved in vitro and in vivo [2]. Such structures are, thus, designated as Non-B-DNA structures or noncanonical nucleic acids. Examples of Non-B-DNA structures are left-handed DNA (Z-DNA), A-DNA (varying in the number of base pairs per turn, twist, and rise), i-motifs, R-loops, hairpins, and cruciforms [3]. It was documented that these structures are more common in living organisms than has been assumed. The presence of these noncanonical structures was confirmed even for plant species [4]. Nowadays, increased scientific effort has been dedicated to understanding the complex role of several noncanonical nucleic acids in plants under suboptimal conditions [5]. On the other hand, the literature dealing with plant G-quadruplexes (G4s) is still very limited, especially in the context of plant stress responses.

G4s are four-stranded DNA structures, which are widely distributed across all three domains of life (Bacteria, Archaea, and Eukaryota) [6,7,8]. The basic building block of G4s—a guanine tetrad—was first described in 1962 by Gellert, Lipsett, and Davies [9], who experimented with guanylic acid and solved its crystal structure. A single guanine tetrad consists of four guanines which are oriented one to another by the angle of 90° and represent one stack of G4. Although it was assumed that mostly two-, three-, and four-stacked G4s are present in the genomes, it was shown, that even five- and six-stacked G4s exist and their formation was confirmed in vitro [10]. Such structure stability can be affected by the metal ions (especially by the monovalent potassium and sodium ions) concentration [11] and their presence within or outside the guanine tetrad (depending on the ion size). Although G4s are currently at the front of interest mainly in cancer research, they have an irreplaceable role in gene expression regulation in general. In plants, both DNA and RNA G4s are documented, and their formation can affect replication [12], transcription [13], and even translation [14] (when the G4 is formed for example in the 5′-UTR of the mRNA [15]). Although in most cases the G4s formation has a repressive effect (suppresses replication, transcription, or translation), recently it was documented that the G4s formation in the coding gene region can have a stimulative effect as well [16]. Regarding gene expression inhibition, G4s act either as a roadblock to RNA polymerases or as a physical obstacle to ribosome entry when located within 5′ UTRs of mRNA [17].

An emerging number of studies dealing with the putative G-quadruplex sites (PQS) prediction in plant genomes appeared, but the number of the in vivo confirmed G4s by their direct detection remains very limited (approximately 1–5% from the predicted PQS—[16]). This fact indicates that the G4s formation can be affected by the outer environmental factors and thus can be much more plastic than was previously assumed. Similar knowledge limitation has to be faced when it comes to our understanding of the connection between G4s formation and stress responses. As the fact that G4s formation is affected by the ion concentrations can be concluded, especially drought, high salinity concentration, or other external factors affecting the concentration of ions available in the cytoplasm should be linked with the major changes in G4 occurrence.

In this study, we decided to search for the PQS in two sets of a well-defined group of genes (the drought stress-responsive genes, and the genes of phenolic compounds synthesis pathways) to find out whether the PQS occurrence significantly varies from the average in the whole *Arabidopsis thaliana* genome and also in comparison with the subset of constitutively expressed (housekeeping) genes. We hypothesize that distinct functional gene sets will have various PQS patterns. Drought stress-responsive genes and phenolic compounds synthesis genes were chosen because these two groups of (at the first sight nonrelated) genes are at the center of our research interest, but nowadays it is evident that drought also influences the synthesis of phenolic compounds [18,19,20]. Therefore, we were interested in whether the PQS patterns of phenolic compounds biosynthesis genes will be similar to the drought stress-responsive genes, or not. Nonetheless, the main aim of this article is to provide a novel methodological approach regarding how to analyze PQS patterns in distinct gene sets. To the best of our knowledge, this is the first study analyzing PQS from such a point of view (in the context of defined gene sets from a single organism).

## 2. Materials and Methods

### 2.1. Data Acquisition and PQS Prediction

FASTA files of the *Arabidopsis thaliana* drought stress-responsive genes (118 genes in total) obtained from the DroughtDB expert-curated database (https://pgsb.helmholtz-muenchen.de/droughtdb/drought_db.html, accessed on 24 July 2022) [21], phenolic compounds biosynthesis genes (30 genes in total), and housekeeping genes (20 genes in total), and FASTA files of their close surroundings 1000 bp upstream (+1000) and downstream (−1000) were acquired from the NCBI database (https://www.ncbi.nlm.nih.gov/, accessed on 30 July 2022). Obtained sequences were processed using our G4Hunter web server (https://bioinformatics.ibp.cz/#/analyse/quadruplex, accessed on 15 August 2022) [22] with the G4Score threshold of 1.2 and window size 25 nt. The annotations of the corresponding chromosome loci were downloaded from the NCBI database as well (where the gene of interest is localized), together with 1000 bp upstream and downstream regions. Other genes and annotated regions overlapped with the genes of interest were manually inspected and removed. For purposes of annotation, only a single isoform per each gene was selected based on the criteria of whether it is a canonical isoform according to UniProt (https://www.uniprot.org/, accessed on 2 September 2022). In case there were more transcripts per one canonical isoform in NCBI, the longer transcript was always chosen. The positions of introns were specified together with 5′-UTRs and 3′-UTRs. Finally, the overlap of corrected/modified annotations with G4Hunter results was calculated.

### 2.2. Data Visualization and Statistical Analysis

A schematic figure of model gene features composition was made using Biorender (https://biorender.com/, accessed on 5 November 2022). Barplots of PQS counts and frequencies within particular NCBI features were made in Microsoft Excel 2021. Statistical analysis of PQS differences between gene sets was performed using the ANOVA test and plotted in GraphPad Prism 9. A heatmap comprising all analyzed genes with non-zero PQS occurrence either within the gene or its close surrounding (±1000 bp) was constructed using the Heatmapper tool [23] (http://www.heatmapper.ca/expression/, accessed on 11 November 2022). The single linkage clustering method and Kendall’s Tau distance measurement method were used to obtain a meaningful number of resulting clusters. All numerical values were normalized by Row Z-score.

## 3. Results

### 3.1. Global Analysis of PQS Occurrence in the Arabidopsis thaliana Genome

In the whole *Arabidopsis thaliana* genome (TAIR 10, comprising five chromosomes, mitochondrial, and plastid DNA), a total of 41,557 PQS were found (Table 1). It may be seen that PQS frequencies per particular genomic chromosomes are very similar in the range of 0.32 to 0.36 PQS per 1000 bp (the mean PQS frequency in genomic DNA was 0.344 PQS per 1000 bp). The highest PQS frequency (1.34 PQS per 1000 bp) was found in mtDNA, which may be partly due to significantly higher GC content (44.79%), than in genomic DNA (around 36%). Plastid DNA contains 0.81 PQS per 1000 bp, which is more than twice as high as genomic DNA with similar GC content. 

### 3.2. Descriptive Analyses of PQS Found within Different Gene Sets and NCBI Features Overlay

The following section summarizes PQS counts and frequencies in three analyzed gene sets (drought stress-responsive genes, phenolic compounds synthesis genes, and housekeeping genes). Even more detailed results are enclosed in Appendix A (PQS counts and frequencies) and Appendix A (NCBI features overlay).

In 118 drought stress-responsive genes analyzed, there were 102 PQS found (gene regions only), or 160 PQS when we are also considering 1000 bp flanks before and after genes. Overall, 35 genes contained exactly one PQS (29.7%). Some genes contained more than one PQS (26 in total, 22%), and some genes had none (a total of 57 genes, 48.3%). The highest number of PQS was found within the 3232 bp long gene HD1 (a total of six PQS). This gene encodes for Histone deacetylase 1, which is an important chromatin-remodeling factor that contributes to transcriptional repression in eukaryotic organisms.

In 30 phenolic compounds synthesis genes analyzed, there were 27 PQS found (gene regions only), or 43 PQS (considering also 1000 bp flanks). Seven genes contained exactly one PQS (23.3%). Six genes contained more than one PQS (20%), and 17 genes had no PQS (56.7%). The highest number of PQS was found within the 1539 bp long gene TT5 (a total of 5 PQS). This gene encodes for the Chalcone-flavanone isomerase family protein which catalyzes the conversion of chalcone into flavanone.

In the set of 20 housekeeping genes analyzed, there were 28 PQS found (gene regions only), or 36 PQS (considering also 1000 bp flanks). Nine genes contained exactly one PQS. Four genes contained more than one PQS (20%), and seven genes had no PQS (35%). The highest number of PQS was found within the 10,519 bp long gene HAF1 (a total of nine PQS). This gene encodes for the HAC13 protein (Histone acetyltransferase of the CBP family 13). The overall results of the percentual distribution of total PQS count in all three gene sets analyzed are depicted in Figure 1. 

Surprisingly, the gene set of housekeeping genes contained the smallest proportion of genes without any PQS, in other words, 65% of housekeeping genes analyzed contained at least one PQS. Next, our analysis revealed that the occurrence of PQS varies notably between distinct genomic features. Because the length of analyzed genes significantly differs and could potentially skew the results, we have always also taken into account a PQS frequency per 1000 bp. Regarding particular gene features (schematically depicted in Figure 2A), the highest PQS frequency per 1000 bp was observed within 5′-UTR regions in phenolic compounds synthesis genes (1.27 PQS per 1000 bp), whereas the lowest PQS frequency per 1000 bp was observed within putative promoter regions of housekeeping genes (0 PQS per 1000 bp), and within intronic regions of phenolic compounds synthesis genes (0 PQS per 1000 bp), then by 0.07 PQS per 1000 bp in drought stress-responsive genes, and 0.08 PQS per 1000 bp in housekeeping genes (Figure 2B–D).

To statistically compare differences in PQS frequencies between particular gene sets we carried out a one-way ANOVA test (separately for gene regions only, and for gene regions with 1000 bp flanks). There were no statistically significant differences between gene sets. However, this may be caused by the small sample size of gene sets, as well as the fact that the data are extremely skewed (many zero “PQS per 1000 bp” values). The distribution of PQS frequencies within particular gene sets is shown in Figure 3. 

Considering particular genes, the highest PQS frequency (up to tenfold compared to the others) was found in the *HRD* gene from the drought-responsive gene set and in the *TT5* gene from the phenolic compounds biosynthesis gene set. The *HRD* gene encodes HRD Integrase-type DNA-binding superfamily protein, and its overexpression increases the density of the roots and improves water and salt stress tolerance in *Arabidopsis thaliana* [24]. These genes could therefore have their regulation dependent on the formation of G4s.

### 3.3. Clustering of PQS Patterns in Analyzed Genes

To inspect PQS patterns in analyzed genes containing at least one PQS, we carried out a cluster analysis based on these four parameters: total PQS counts in the whole gene regions; total PQS counts in gene regions with 1000 bp flanks on both ends; PQS frequencies per 1000 bp in the whole genes; and PQS frequencies in gene regions with 1000 bp flanks on both ends. As a result, five main clusters were obtained (Figure 4). Cluster I (the biggest one) contained 46 genes, cluster II contained 7 genes, cluster III contained 14 genes, cluster IV contained 24 genes, and cluster V (the smallest one) contained 3 genes (out of 111 genes containing at least 1 PQS either in gene regions or within 1000 bp flanks). Noticeably, in cluster V there were only drought-responsive genes with high PQS frequencies in gene regions, but not in 1000 bp flanks, where PQS frequencies were low. On the other hand, cluster IV gathered genes with relatively high PQS frequencies when also considering 1000 bp flanks, but low PQS frequencies in gene regions themselves. Clusters III and II were quite similar and had low PQS frequencies both within gene regions as well as 1000 bp flanks. The largest cluster I contained genes with high total PQS counts, variable PQS frequencies in gene regions, and low PQS frequencies in 1000 bp flanks. Finally, it can be seen that each of the analyzed gene sets contained both “PQS-rich” and “PQS-poor” genes.

## 4. Discussion

Our previously published approach for PQS prediction showed that they are nonrandomly distributed in the *Pisum sativum* genome [25], where the frequency of PQSs for nuclear DNA was 0.42 PQSs per 1000 bp in nuclear DNA, 0.53 PQS per 1000 bp in plastid DNA, and 1.58 PQS per 1000 bp in mitochondrial DNA. A similar trend is observable also in current data from *Arabidopsis thaliana*, where the frequency of 0.34 PQS per 1000 bp was found in nuclear DNA, a higher frequency of 0.81 in plastid DNA, and the highest frequency of 1.34 in mitochondrial DNA. This result indicates the importance of PQS in circular and especially mitochondrial genomes. While in algae and fungi the frequency of PQS in mtDNA is lower than in plants, including *Arabidopsis thaliana*, in vertebrates the frequency of PQS in mtDNA is even significantly higher [26]. Although such large-scale predictions are computationally extensive, it would be beneficial to also inspect other higher plants in the future to see if this trend (frequency of PQS in nuclear DNA < frequency of PQS in plastid DNA < frequency of PQS in mitochondrial DNA) is always the same. These data should then be correlated with GC content as well because generally higher GC content may favor more PQS, although recently it was shown that genomes with a very similar GC content may differ significantly in their PQS frequencies [27]. Considering nuclear and plastid genomes which have very similar GC content in *Arabidopsis thaliana,* an explanation for different PQS frequencies may be found in significantly different gene densities. In the nuclear genome, there are 31.8 genes per 100,000 bp, whereas in the plastid genome, more than twice the genes (83.5 per 100,000 bp) are present (unpublished data, computation is based on the information from the NCBI database, https://www.ncbi.nlm.nih.gov/data-hub/genome/GCF_000001735.4/, accessed on 28 December 2022). Higher PQS frequencies could therefore positively correlate with higher gene density in the plastid genome because each gene is a potential source of PQS-rich regulatory features (e.g., 5′-UTRs). In addition, this article has shown that the PQS frequencies within the gene regions were generally higher than the average PQS frequency of the whole *Arabidopsis thaliana* genome (comprising both genic and intergenic loci).

Our recent analysis in *Arabidopsis thaliana* further revealed that both the total number and frequencies of PQS within introns were very low (at least in three gene sets analyzed in this study), as only 10 PQS were found in these locations in total (8 in drought-responsive genes, 2 in housekeeping genes, and 0 in phenolic compounds synthesis genes). On the other hand, PQS frequencies within CDS were much higher. This fact further strengthens the hypothesis about the regulatory potential of not only DNA G4s, but also RNA G4s which may be formed in mRNA prior to or during the translation (PQS-poor introns are excised during the mRNA processing by the spliceosome complex). It is worth noting that the mean and median lengths of introns are only 168 bp, or 100 bp, respectively, in *Arabidopsis thaliana* (for comparison, in maize, the mean intron length is 516 bp and the median is 146 bp; and in humans, the mean intron length is 3356 bp and the median is 1023 bp) [28]. With such small intron sizes in *Arabidopsis thaliana*, PQS (usually 25 bp long) would span a substantial part of them, and maybe this is one of the reasons why they are depleted there—to avoid potential problems with RNA splicing.

A similar trend as in the case of introns was observed in the regions 1000 bp downstream of genes (end_1000)—both absolute and relative PQS frequencies were very low. Such an observation would make sense, as the regulation behind the gene region would not be necessary compared to the need for precise regulation in the promoter sequence of the same gene. Nonetheless, in further studies, a whole genome of (not only) *Arabidopsis thaliana* could be analyzed by individual genes/gene sets and their features to gain an even more comprehensive picture of PQS distribution within functional units/pathways of genome and transcriptome.

One of the most interesting things found in this study is the fact that in all three gene sets analyzed, there were always slightly higher PQS frequencies in CDS than in the whole *Arabidopsis thaliana* genome (mean PQS frequency). Traditionally it has been assumed that the frequency of PQS in noncoding regions, such as promoters, telomeres [29], and untranslated regions [30], is much higher than within coding regions. Nonetheless, additional data are needed to possibly generalize this observation for the plants (and other eukaryotic genomes).

In plants, there is still very little evidence about the mechanistic function of particular PQS and corresponding G4. One of the first studies dealing with such problematics was published by Cho et al. [14], focusing on particular PQS in 5′-UTR regions of *SMXL4* and *SMXL5* genes in *Arabidopsis thaliana*. These genes are responsible for phloem differentiation. It was observed that the formation of G4s within 5′-UTR of mRNA molecule (induced/stabilized by G4-binding protein JULGI) causes suppression of SMXL4/5 translation and restricts phloem differentiation [14]. Another work focused on a single PQS was published by Kwok et al. [31], where they identified PQS and corresponding G4 within the 5′-UTR of the *ATR* gene in *Arabidopsis thaliana*. Transient reporter gene assay further revealed that this G4 inhibits translation, but not the transcription of the *ATR* gene in living plants [31]. G4s also have potential in applied/agronomic science, as it is believed that their formation and stability (and thereby the expression of genes containing PQS) could be affected by environmental conditions, e.g., macronutrient availability (mainly K^+^ and Na^+^ ions) and/or drought [30], temperature [32], irradiation [33], etc. From this point of view, it would be interesting to know which genes and pathways contain the highest amount/frequencies of PQS. All in all, we believe that this study and used approach will inspire scientists in the G4 field to gradually answer more and more remaining questions about the integral map of PQS within (not only) plant genomes.

## Figures and Tables

**Figure 1 life-13-00199-f001:**
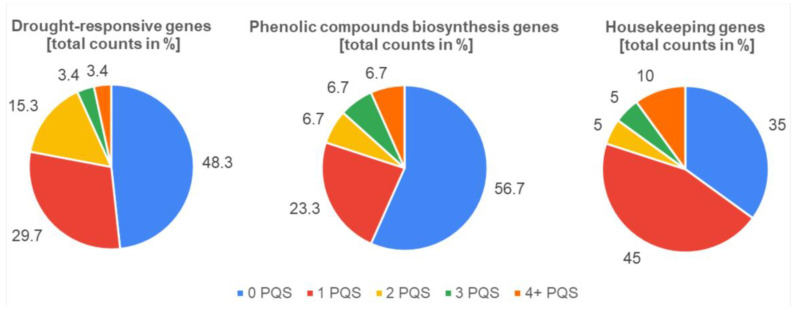
Percentual distribution of total PQS counts in different gene sets. All analyzed genes were divided into five categories according to their total PQS counts (zero, one, two, three, four, and more PQS).

**Figure 2 life-13-00199-f002:**
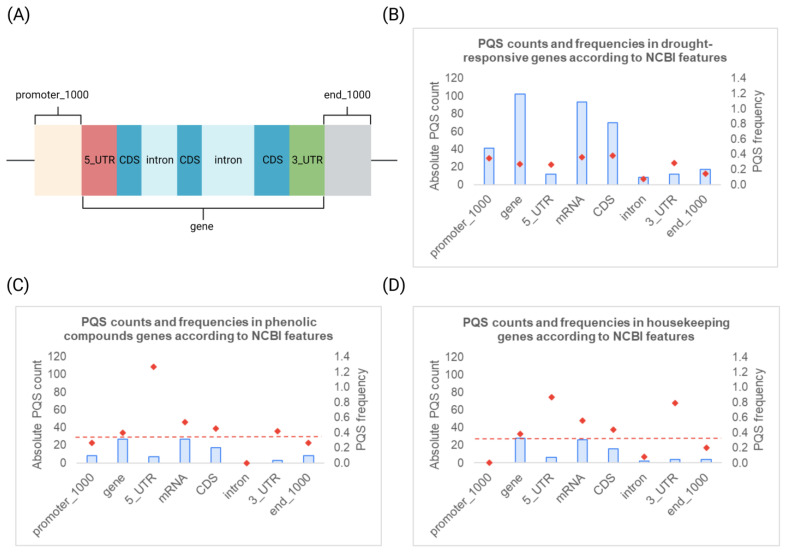
Analyses of PQS frequencies within NCBI features. (**A**) Scheme of NCBI features used in this study to categorize DNA regions for PQS analysis. (**B**) PQS counts and relative frequencies in drought-responsive genes. (**C**) PQS counts and relative frequencies in phenolic compounds synthesis genes. (**D**) PQS counts and relative frequencies in selected housekeeping genes. Blue bars express the total PQS counts found in particular NCBI features, whereas red marks depict relative PQS frequencies (PQS count per 1000 bp). Red dashed horizontal lines indicate the mean PQS frequency (0.344 PQS per 1000 bp) in the *Arabidopsis thaliana* genome.

**Figure 3 life-13-00199-f003:**
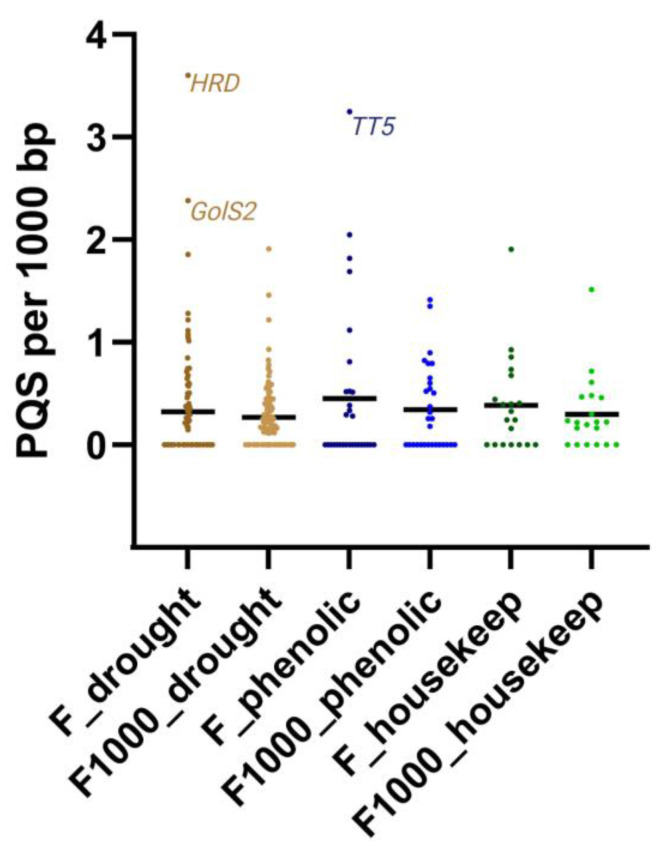
Gene sets comparison. Distributions of all “PQS per 1000 bp” values in analyzed gene sets are shown. F stands for PQS frequencies within gene regions only, and F1000 stands for PQS frequencies in genes with 1000 bp flanks on both sides. Black horizontal lines indicate mean PQS frequency values in particular gene sets, and three outliers/genes with the highest PQS frequencies are depicted.

**Figure 4 life-13-00199-f004:**
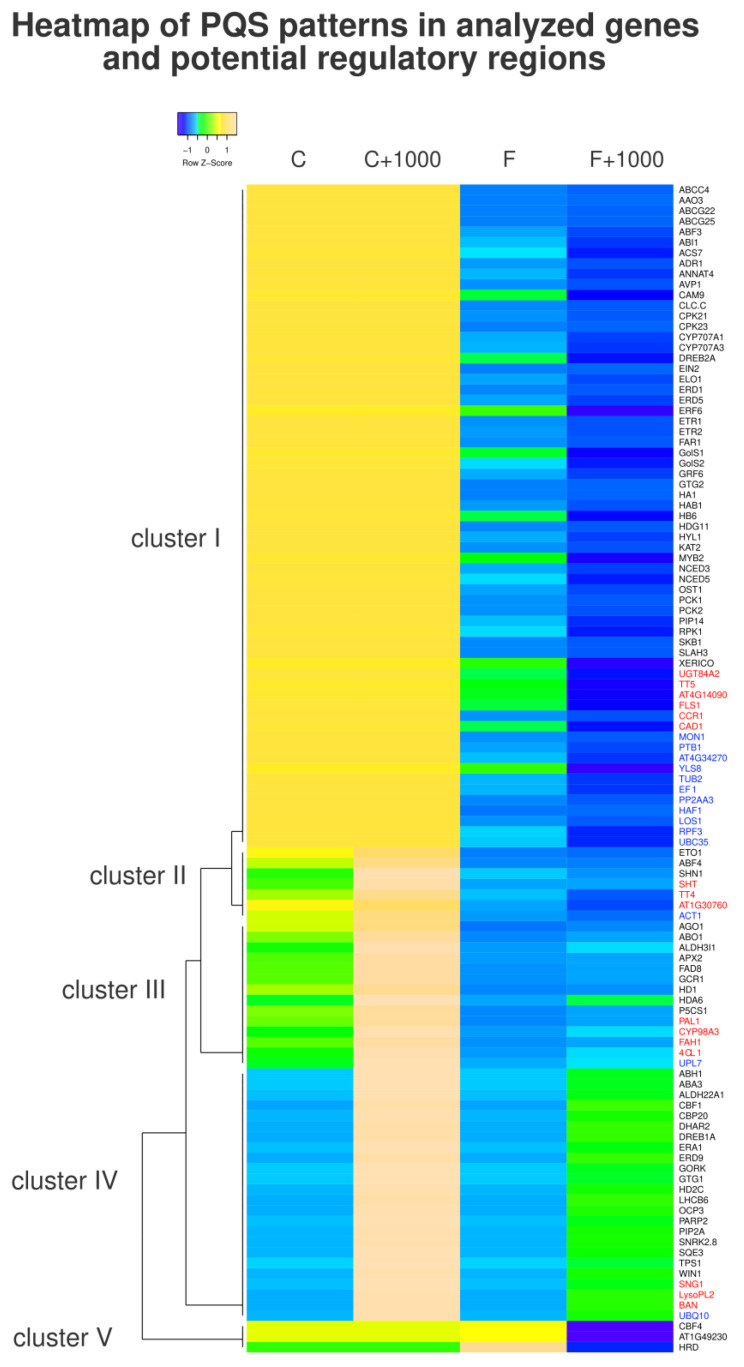
Heatmap comprising all genes with non-zero PQS occurrence either within the gene or its close surrounding (±1000 bp) analyzed in this study. The column designated C is based on absolute PQS counts within gene regions, and column F is based on PQS frequencies per 1000 bp within gene regions. Columns C+1000 and F+1000 are focused on PQS counts and frequencies within the close surrounding of genes (±1000 bp). Five resulting clusters of genes are designated on the left side. The single linkage clustering method and Kendall’s Tau distance measurement method were applied, and all numerical values were normalized by Row Z-score. Drought-responsive gene names are in black, phenolic compounds biosynthesis gene names are in red, and housekeeping gene names are in blue.

**Table 1 life-13-00199-t001:** PQS counts and frequencies in the whole *Arabidopsis thaliana* genome.

Chromosome	NCBI ID	Length	GC Content	PQS Count	Frequency per 1000 bp
chr1	NC_003070.9	30,427,671	35.68	10,989	0.36
chr2	NC_003071.7	19,698,289	35.86	6613	0.34
chr3	NC_003074.8	23,459,830	36.32	7893	0.34
chr4	NC_003075.7	18,585,056	36.20	6781	0.36
chr5	NC_003076.8	26,975,502	35.93	8664	0.32
mtDNA	NC_037304.1	367,808	44.79	492	1.34
cpDNA	NC_000932.1	154,478	36.29	125	0.81

## Data Availability

The data presented in this study are available in this article and Appendix A.

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
