# Peer review of "Analysis of G-Quadruplex-Forming Sequences in Drought Stress-Responsive Genes, and Synthesis Genes of Phenolic Compounds in Arabidopsis thaliana"

_life, 2023, doi:10.3390/life13010199_

Round 1

Reviewer 1 Report

The manuscript by Pečinka et al. carried out a case study of putative G-quadruplex sites in drought stress-responsive genes, and genes related to the phenolic compounds biosynthesis in Arabidopsis thaliana. The authors have analyzed % distribution of total PQS counts, frequencies and distribution patterns in different gene sets. The results presented here are informative and relevant for the 'life.' However, the data presented here are premature for publication at this stage. The authors need to address some critical concerns in the current version of the manuscript before its publication. The issues are listed below.

The Abstract is written in very casual language. It needs to be rewritten. 

The rationale behind choosing phenolic compounds biosynthesis genes, and housekeeping genes is underdeveloped. Please explain precisely as to why the authors decide to study these genes. 

Can the authors comment on why plastid DNA contains 0.81 PQS per 1000 bp? 

The authors claim that ‘These genes could therefore have their regulation dependent on the formation of G quadruplexes.’ However, no substantial evidence is provided to support this claim. 

Although the authors have made an interesting discovery about the PQS counts, they did not provide convincing arguments to their functional requirements, either computationally, or experimentally. 

There are many verb tense inconsistencies throughout the manuscript. Further, the writing style, typographical and grammatical errors should be corrected in the revised version of the manuscript.

Author Response

Dear Reviewer, many thanks for your kind recommendation. We appreciate all of your suggestions and time devoted to reviewing our article. We did our best to incorporate appropriate changes into the manuscript (tracked changes mode). Please find our answers in bold below.

The Abstract is written in very casual language. It needs to be rewritten.

Thank you for this comment. We double-checked the Abstract and make some changes. However, in the Life journal, there is also a Simple summary (except for the Abstract), where the casual language is intended to allow a large audience to understand the article content in a nutshell.

The rationale behind choosing phenolic compounds biosynthesis genes, and housekeeping genes is underdeveloped. Please explain precisely as to why the authors decide to study these genes.

Thank you for this suggestion, we extended the rationale behind the selection of all three gene sets used in this study. Please, see the newest version of the manuscript.

Can the authors comment on why plastid DNA contains 0.81 PQS per 1000 bp?

Thank you for this suggestion, we have added a possible explanation to the revised version of the manuscript. In summary, we think that this is mainly due to the high frequency of genes (and their regulatory regions such as 5-UTRs) localized in plastid DNA. In fact, there are around 32 genes per 100,000 bp in genomic DNA, versus 84 genes per 100,000 bp in plastid DNA. Such an explanation would be in good congruence with our limited data, where the PQS content within the genic regions of all three gene sets analyzed was always higher than the average PQS content of the whole Arabidopsis thaliana genome (comprising both genic and intergenic regions).

The authors claim that ‘These genes could therefore have their regulation dependent on the formation of G quadruplexes.’ However, no substantial evidence is provided to support this claim. Although the authors have made an interesting discovery about the PQS counts, they did not provide convincing arguments to their functional requirements, either computationally, or experimentally.

Thank you for these comments. This study aimed to carry out a descriptive bioinformatic analysis of PQS counts and occurrence within defined gene sets, which could serve as a template/inspiration for further studies dealing with PQS analyses. Validation of the regulatory potential of a particular PQS/G-quadruplex structure is usually done via reporter assay in vivo, in plant models, this is not a trivial task and was done only in 2 studies so far, for SMXL4/5 and ATR genes. We have added this information/references to the Discussion section

There are many verb tense inconsistencies throughout the manuscript. Further, the writing style, typographical and grammatical errors should be corrected in the revised version of the manuscript.

Thank you for the suggestion, unfortunately, none of us is a native English speaker. We double-checked the whole manuscript again and made some improvements. We believe we will further fix this issue during the Proofreading phase, together with the kind help of journal typesetters and language specialists.

Reviewer 2 Report

This study provided some meaningful information related to the potential regulated function of G-quadruplex in plant response to drought stress. However, I have some concerns.

1. What criteria did you use to screen those 118 drought-responsive genes from the Arabidopsis genome in this study? 

2. Please explain in detail why you perform an analysis of the PQS in the genes related to phenolic compound synthesis in the introduction.  

Author Response

Dear Reviewer, many thanks for your kind recommendation. We appreciate all of your suggestions and did our best to incorporate appropriate changes into the manuscript (tracked changes mode). Please find our answers in bold below.

  1. What criteria did you use to screen those 118 drought-responsive genes from the Arabidopsis genome in this study?

Thank you for this question, we have used the DroughtDB: Drought Stress Gene Database (https://pgsb.helmholtz-muenchen.de/droughtdb/drought_db.html) as the main source of information about drought stress-related genes. We extended chapter 2.1 in the manuscript and added an appropriate quotation of the DroughtDB paper (https://academic.oup.com/database/article/doi/10.1093/database/bav046/2433179).

  1. Please explain in detail why you perform an analysis of the PQS in the genes related to phenolic compound synthesis in the introduction.  

Thank you for this suggestion, we extended the rationale behind the selection of all three different gene sets used in this study. Please, see the newest version of the manuscript.

Author Response

Dear Reviewer, many thanks for your kind recommendation. We appreciate all of your suggestions and did our best to incorporate appropriate changes into the manuscript. Please find our answers in bold below.

1, The author should give more information or examples about the function of G-quadruplex-forming sequences in the introduction or discussion section.

Thank you for this suggestion, we have added specific examples of the function of G-quadruplex-forming sequences within particular plant genes (SMXL4/5 and ATR). Please see the Discussion section (tracked changes).

 2, The relationship between the number of PQS and gene expression should be discussed and more evidence should be conducted.

Thank you for this comment. Unfortunately, there is no direct relationship between the number of PQS (or PQS frequency) and gene expression. However, intuitively more PQS indicate more regulatory potential via G4s. In general, PQS can give rise to a stable G-quadruplex structure(s), which may inhibit gene expression, either on the transcription (G4 in DNA) or translation levels (G4 in mRNA). Nonetheless, the level of inhibition depends also on additional factors influencing G4 stability, such as the presence of G4-binding proteins and ions. On the other hand, our current paper aims to propose a novel approach on how to analyze PQS counts and frequencies within defined gene sets, that (to the best of our knowledge) wasn’t applied so far. We believe this approach will be useful to identify PQS/G4-enriched and PQS/G4-poor gene sets/pathways and this information will give us further clues about potential G4 function(s) and evolution. Nonetheless, this will take much computational time and ideally should be done for all plant genomes and released in the form of a user-friendly online database. We are not able to do that now due to computational limitations, but we believe that many people worldwide will get inspired and will continue in our effort, possibly not only in Plant Science but also with cataloging PQS in animals, fungi, prokaryotes, and viruses (according to their functional gene sets/pathways).

 3, How the PQS regulate the gene expression?

Thank you for the comment, PQS can give rise to four-stranded G4 structures which form either a roadblock to RNA polymerases, or a physical obstacle to ribosome entry when located within 5′ UTRs of mRNA. We have added this information to the Introduction section, together with references, please see the revised manuscript in Tracked changes mode.

4, Many public data for Arabidopsis thaliana under different stress can be used, I think the author should analyze more stress genes such as salt response and cold response genes and so on.

Thank you for your suggestion. This article was intended mainly as a methodological paper/case study, without inferring biological phenomena (this would require much computationally intensive mode and analyses of all gene sets/pathways, at least in Arabidopsis thaliana). We are not able to do that at this time, but we believe it will stimulate further research efforts in the field.

Round 2

Reviewer 1 Report

After careful examination of the revised manuscript, the response of the authors to previous reviews, and the changes made in the manuscript, I gather that the revised version of the manuscript has addressed the major concerns raised in the previous version of the paper (the limitations about unresolved comments are understandable). Hence, I endorse the publication of this paper.

Reviewer 3 Report

Dear editors, this paper can be accept.